# Foot self-care practices among adults with diabetes in the West Bank, Palestine: A cross-sectional study

Alhareth M. Amro[1], Salahaldeen Deeb[1]*, Mohammad Harbi Alfrookh[1], Bushra M. Makhamra[2], Lama Amro[2], Anas K. Assi[1], Osama J. Makhamreh[1], Anas Ishqair[1], Afnan W. M. Jobran[1]

**1** Faculty of Medicine, Al-Quds University, Jerusalem, Palestine, **2** Palestine ministry of Health, Hebron, Palestine

* salahdeeb2001@gmail.com

## Abstract

Diabetic foot complications are a major cause of morbidity and disability yet are largely preventable through proper self-care. This cross-sectional study assessed foot self-care practices among adults with diabetes in the West Bank, Palestine, and identified factors associated with adherence. We interviewed 300 adults attending diabetes clinics at four sites using a structured questionnaire adapted from a published foot-care instrument. Foot self-care was measured using an 11-item frequency scale (score range 0–44), with higher scores indicating better adherence; the scale showed acceptable internal consistency (Cronbach's alpha = 0.81). Overall adherence was suboptimal: although 80.9% reported washing their feet daily, only 33.1% inspected their feet daily, 45.1% dried between toes, and 40.3% inspected shoes. Risky behaviors were common, including soaking feet (28.6%), walking barefoot (20.2%), and wearing shoes without socks (15.9%). Only 13% reported performing all recommended behaviors daily. In multivariable analysis, higher income, higher educational attainment, and longer diabetes duration were associated with better adherence, and male sex was associated with lower adherence in the ordinal logistic model. These findings highlight important gaps in preventive foot self-care in the West Bank, driven by socioeconomic and educational disparities and limited clinical reinforcement, underscoring the need for culturally tailored education and routine provider-led counseling and screening to reduce diabetic foot complications.

## Introduction

Diabetes mellitus (DM) is a heterogeneous metabolic disorder characterized by persistent hyperglycemia and glucose intolerance, resulting from either insulin deficiency, impaired insulin action, or a combination of both [1]. It is among the most

**Data availability statement:** The dataset contains human participant clinical and sociodemographic information. Although de-identification was performed, public sharing may still pose a risk of participant re-identification due to the combination of variables and the study context. Therefore, the data are not publicly available. De-identified data are available upon reasonable request to qualified researchers, subject to approval by the Institutional Review Board (IRB), Al-Quds University, and completion of a data-use agreement. Requests should be sent to: Research Office, Al-Quds University (Research@admin.alquds.edu).

**Funding:** The authors received no specific funding for this work.

**Competing interests:** The authors have declared that no competing interests exist.

prevalent metabolic diseases worldwide, currently affecting an estimated 371 million individuals [2]. According to the World Health Organization (WHO), by 2025 nearly 60% of the global diabetic population will reside in developing countries, particularly across Asia [3].

Diabetes is associated with a wide range of acute and chronic complications, including coronary artery disease, nephropathy, retinopathy, microalbuminuria, and neuropathy. Among these, diabetic neuropathy is one of the most frequent and debilitating complications. Importantly, foot-related complications in diabetic patients remain some of the most preventable outcomes of the disease [3].

A diabetic foot ulcer (DFU) is defined as a full-thickness wound that penetrates through the dermis and commonly occurs below the ankle in individuals with diabetes. If not properly treated, these ulcers can progress to infection and severe sequelae [4]. The annual incidence of new foot ulcers among diabetic patients is approximately 2.2%, rising to 5.8% within three years. Over a lifetime, about 25% of diabetic patients are at risk of developing a foot ulcer, with 10–15% eventually experiencing this complication. Alarmingly, 5–24% of DFU cases may progress to the point of requiring limb amputation [3].

Effective diabetes management plays a central role in preventing both acute and chronic complications. Acute complications can often be avoided, while the risk of long-term complications can be significantly reduced through patient education and consistent monitoring of blood glucose levels. Preventive strategies include encouraging regular physical activity, weight control, adherence to a balanced diet, and maintaining strict glycemic control [5]. Moreover, limited knowledge and poor foot care practices among diabetic patients are recognized as significant risk factors for the development of foot complications. Therefore, providing timely education on foot self-care is essential to minimize or prevent diabetic foot problems [3]. The present study examines foot self-care practices among adults with diabetes in the West Bank, Palestine, and evaluates factors associated with better preventive foot-care behaviors. The findings will help guide culturally appropriate education and clinic-based prevention strategies in the Palestinian context.

## Methodology

### Study design and setting

We conducted a cross-sectional survey of adults with diabetes in Palestine. Data collection took place from 15/01/2025–19/09/2025 at four sites that provide routine diabetes care: Princess Alia Governmental Hospital, Palestinian Medical Complex, Dura Governmental Hospital, and Al Ahli Hospital. Interviews were conducted on site in private areas to ensure confidentiality and comfort.

The four study sites were purposively selected to capture variation in diabetes care delivery across the West Bank and to support consistent implementation of study procedures. The selected governmental hospitals, medical complex, and non-governmental hospital are major diabetes-care providers and were chosen based on patient volume, availability of clinic days, administrative approval, and adequate

space for private interviews. The sites were also conveniently reachable for the data-collection team, which enabled repeated visits and uniform recruitment procedures across all sites.

During routine diabetes clinic days at each participating center, approximately 50 adults with diabetes attend per day (≈200 patients/day across the four sites). Routine care typically includes follow-up for glycemic monitoring, medication refills, and complication screening; however, the amount of structured foot-care counseling may vary by provider and site. To improve representativeness and reduce day-specific selection effects, recruitment was conducted on multiple clinic days across sites and all eligible attendees present during data-collection days were approached consecutively.

## Sampling and sample size

We used consecutive sampling of eligible adults attending routine diabetes visits during the data-collection period. Data collection occurred on clinic days agreed with each site, and all eligible patients present during those periods were approached in sequence to minimize selection based on patient characteristics.

The target sample size was set at n = 300 to estimate the prevalence of key preventive foot-care behaviors with acceptable precision. Assuming a conservative expected prevalence of 50% (maximizing variance), 95% confidence level (Z = 1.96), and a 6% absolute margin of error, the required sample was approximately 267 participants; we increased this target to 300 to improve precision and accommodate incomplete responses.

## Participants

Eligible participants were adults (≥18 years) with a clinician diagnosis of diabetes mellitus (type 1 or type 2) for at least 6 months, attending routine diabetes care visits at one of the participating facilities during the study period, and able to complete an interview in Arabic. We excluded individuals who were acutely ill at the time of the visit, had cognitive impairment or severe hearing/speech limitations preventing informed consent or interview completion, or declined participation. Recruitment used a consecutive sampling approach in clinic waiting areas on pre-specified data-collection days. Trained interviewers confirmed eligibility, obtained consent, and completed the interview during the same visit.

Potential participants were approached face-to-face in the waiting area before or after their routine visit; no telephone or post-visit contact was used. Data were collected by trained medical students and clinicians, supervised by the study investigators, using standardized interviewer training and scripts.

## Questionnaire development and translation

The interview used a structured questionnaire adapted from a previously published diabetic foot-care instrument developed and pilot-tested in Oman [6]. The questionnaire included: (1) sociodemographic variables (e.g., age, sex, residence, education, employment, income); (2) diabetes-related clinical history (type of diabetes, duration, treatments, self-reported complications, prior ulcer/amputation, and foot deformities); (3) health-care use (clinic visits, foot examinations, exposure to foot-care education topics); and (4) foot self-care behaviors assessed using an 11-item frequency scale (see "Outcome measures and scoring").

The original instrument was reviewed by the study team for contextual relevance. The final interview was administered in Arabic. Items were translated into Arabic and reviewed for clarity by bilingual clinicians; minor wording adjustments were made to ensure comprehension in the Palestinian context. Interviewers were trained to follow a standardized script and avoid leading prompts. At the end of each interview, responses were reviewed with the participant for completeness.

## Scoring and reliability

Foot self-care was assessed with eleven behaviors that covered inspection, hygiene, thermal safety, shoe inspection, moisturizing product use, and behaviors that increase risk. Responses were harmonized to a five-level numeric scale

where never equaled 0, rarely equaled 1, once a month equaled 2, once a week equaled 3, and daily equaled 4. The four negatively worded behaviors-soaked feet, walked barefoot, wore shoes without socks, and cut own toenails were reverse scored using 4 minus the original score so that higher values indicated better self-care. Item scores were summed to create a total score that ranged from 0 to 44. Adherence categories were defined as very poor 0–11, poor 12–22, average 23–33, and good 34–44. Internal consistency of the eleven-item scale was planned to be evaluated with Cronbach alpha and item total correlations, with alpha if item dropped as a secondary check.

Internal consistency of the 11-item scale in this sample was acceptable (Cronbach's alpha = 0.81).

## Data management and statistical analysis

Interviewers checked forms for completeness before ending each encounter. Data were entered into a structured workbook and verified with range and logic checks. We summarized missingness for every variable. For the foot self-care items, if one or two items were missing for a participant we imputed those values with the item median across the sample. If three or more items were missing, we set the total score to missing for that participant. Continuous clinical variables were screened for outliers using Tukey fences and key analyses were repeated after excluding flagged values.

Descriptive statistics included means and standard deviations and medians and interquartile ranges for continuous variables and counts and percentages for categorical variables. Normality of the total score was examined with the Shapiro Wilk test together with histograms, kernel density plots, and inspection of skewness and kurtosis. Group comparisons of the total score used Welch t tests for two level variables and analysis of variance for multi-level variables when distributions were approximately normal, and the Mann Whitney U test or Kruskal Wallis test when clearly non normal. Associations with continuous predictors used Pearson and Spearman correlations, using Spearman as primary if normality was rejected. The primary adjusted analysis was ordinary least squares linear regression with robust standard errors and prespecified predictors that included age, sex, residence, education, income, employment, smoking, diabetes type, insulin use, duration of diabetes, glycated hemoglobin when available, neuropathy, prior foot ulcer, prior amputation, foot deformity, and number of clinic visits in the prior year. We examined residual diagnostics for the final OLS model (normality and homoscedasticity) and retained robust standard errors to account for any heteroscedasticity. A secondary model used ordinal logistic regression for the four-level adherence outcome with checks of the proportional odds assumption and alternative models if the assumption was not met. Multiple testing in bivariable analyses was controlled using the Benjamini Hochberg false discovery rate at 0.05. Sensitivity analyses repeated primary models using complete cases and after excluding outliers. Data processing and analysis were performed in Python using pandas, numpy, scipy, statsmodels, and matplotlib.

## Ethical approval

**The study protocol, survey instrument, and verbal consent procedure were approved by the Institutional Review Board of Al-Quds University (Ref: 477/REC/2025). Participation was voluntary and confidential. Before the interview, trained data collectors read an IRB-approved Arabic consent script describing the study purpose, procedures, risks/benefits, confidentiality, and the right to refuse or withdraw at any time. Participants who agreed provided verbal informed consent, which was documented by the interviewer on a consent log (date/time and participant study ID) and witnessed by a clinic staff member or a second member of the data-collection team.**

## Results

### Participant characteristics

A total of 300 adults with diabetes were included. The sample was 54.7% female (164 women and 136 men) with a mean age of approximately 55 years (Table 1). No participants were excluded for missing demographic data. The most common

**Table 1. Participant characteristics overall and by foot self-care adherence category.**

| Characteristic | | Overall (N = 300) | Very poor (N = 1) | Poor (N = 85) | Average (N = 171) | Good (N = 39) | p-value |
|---|---|---|---|---|---|---|---|
| Sex | Female | 164 (54.7%) | 0 (0.0%) | 49 (57.6%) | 88 (51.5%) | 25 (64.1%) | 0.65 |
| | Male | 136 (45.3%) | 1 (100.0%) | 36 (42.4%) | 83 (48.5%) | 14 (35.9%) | |
| Diabetes type | Type 1 | 72 (24.0%) | 0 (0.0%) | 23 (27.1%) | 36 (21.1%) | 10 (25.6%) | 0.604 |
| | Type 2 | 228 (76.0%) | 1 (100.0%) | 62 (72.9%) | 135 (78.9%) | 29 (74.4%) | |
| Current diabetes treatment | Diet alone | 12 (4.0%) | 0 (0.0%) | 3 (3.5%) | 4 (2.3%) | 4 (10.3%) | |
| | Oral agents only | 148 (49.3%) | 1 (100.0%) | 46 (54.1%) | 77 (45.0%) | 21 (53.8%) | |
| | Insulin | 63 (21.0%) | 0 (0.0%) | 7 (8.2%) | 48 (28.1%) | 8 (20.5%) | |
| | Oral agents and insulin | 77 (25.7%) | 0 (0.0%) | 29 (34.1%) | 42 (24.6%) | 6 (15.4%) | |
| Age group | ≤19 | 5 (1.7%) | 0 (0.0%) | 1 (1.2%) | 2 (1.2%) | 2 (5.1%) | 0.862 |
| | 20–29 | 18 (6.0%) | 0 (0.0%) | 4 (4.7%) | 9 (5.3%) | 4 (10.3%) | |
| | 30–39 | 10 (3.3%) | 0 (0.0%) | 2 (2.4%) | 5 (2.9%) | 3 (7.7%) | |
| | 40–49 | 30 (10.0%) | 0 (0.0%) | 10 (11.8%) | 15 (8.8%) | 5 (12.8%) | |
| | 50–59 | 92 (30.7%) | 0 (0.0%) | 25 (29.4%) | 55 (32.2%) | 12 (30.8%) | |
| | 60–69 | 96 (32.0%) | 1 (100.0%) | 29 (34.1%) | 56 (32.7%) | 10 (25.6%) | |
| | ≥70 | 49 (16.3%) | 0 (0.0%) | 14 (16.5%) | 29 (17.0%) | 6 (15.4%) | |
| Highest education | None | 26 (8.7%) | 0 (0.0%) | 8 (9.4%) | 16 (9.4%) | 2 (5.1%) | 0.046 |
| | Some primary (grade 1–6) | 47 (15.7%) | 0 (0.0%) | 19 (22.4%) | 23 (13.5%) | 5 (12.8%) | |
| | Some secondary (grade 7–9) | 35 (11.7%) | 0 (0.0%) | 9 (10.6%) | 21 (12.3%) | 5 (12.8%) | |
| | Some high school (grade 10–12) | 35 (11.7%) | 0 (0.0%) | 7 (8.2%) | 21 (12.3%) | 7 (17.9%) | |
| | Completed high school | 44 (14.7%) | 0 (0.0%) | 12 (14.1%) | 25 (14.6%) | 7 (17.9%) | |
| | Vocational degree | 12 (4.0%) | 0 (0.0%) | 3 (3.5%) | 6 (3.5%) | 3 (7.7%) | |
| | Undergraduate university | 65 (21.7%) | 1 (100.0%) | 18 (21.2%) | 36 (21.1%) | 10 (25.6%) | |
| | Postgraduate university | 6 (2.0%) | 0 (0.0%) | 1 (1.2%) | 4 (2.3%) | 1 (2.6%) | |
| Tobacco use | Yes (current) | 66 (22.0%) | 0 (0.0%) | 18 (21.2%) | 35 (20.5%) | 13 (33.3%) | 0.067 |
| | No (never) | 191 (63.7%) | 1 (100.0%) | 54 (63.5%) | 111 (64.9%) | 25 (64.1%) | |
| | Former | 43 (14.3%) | 0 (0.0%) | 13 (15.3%) | 20 (11.7%) | 1 (2.6%) | |
| Any other chronic disease: | Yes | 257 (85.7%) | 1 (100.0%) | 71 (83.5%) | 148 (86.5%) | 37 (94.9%) | 0.802 |
| | No | 43 (14.3%) | 0 (0.0%) | 14 (16.5%) | 23 (13.5%) | 2 (5.1%) | |
| Neuropathy diagnosed | Yes | 118 (39.3%) | 0 (0.0%) | 33 (38.8%) | 71 (41.5%) | 14 (35.9%) | 0.745 |
| | No | 121 (40.3%) | 1 (100.0%) | 36 (42.4%) | 68 (39.8%) | 16 (41.0%) | |
| | Unknown | 61 (20.3%) | 0 (0.0%) | 16 (18.8%) | 32 (18.7%) | 13 (33.3%) | |
| History of foot ulcer | Yes | 56 (18.7%) | 0 (0.0%) | 13 (15.3%) | 34 (19.9%) | 9 (23.1%) | 0.052 |
| | No | 234 (78.0%) | 1 (100.0%) | 70 (82.4%) | 133 (77.8%) | 30 (76.9%) | |
| | Unsure | 10 (3.3%) | 0 (0.0%) | 2 (2.4%) | 4 (2.3%) | 0 (0.0%) | |
| History of amputation | Yes | 20 (6.7%) | 0 (0.0%) | 4 (4.7%) | 13 (7.6%) | 3 (7.7%) | 0.057 |
| | No | 275 (91.7%) | 1 (100.0%) | 80 (94.1%) | 156 (91.2%) | 38 (97.4%) | |
| | Unsure | 5 (1.7%) | 0 (0.0%) | 1 (1.2%) | 2 (1.2%) | 0 (0.0%) | |
| Foot deformity present | Yes | 130 (43.3%) | 0 (0.0%) | 37 (43.5%) | 80 (46.8%) | 13 (33.3%) | 0.803 |
| | No | 153 (51.0%) | 1 (100.0%) | 44 (51.8%) | 85 (49.7%) | 23 (59.0%) | |
| | Unknown | 17 (5.7%) | 0 (0.0%) | 4 (4.7%) | 6 (3.5%) | 3 (7.7%) | |
| Can reach and see soles of feet | Yes | 202 (67.3%) | 0 (0.0%) | 56 (65.9%) | 120 (70.2%) | 26 (66.7%) | – |
| | No | 86 (28.7%) | 1 (100.0%) | 25 (29.4%) | 45 (26.3%) | 9 (23.1%) | |
| | Unsure | 12 (4.0%) | 0 (0.0%) | 4 (4.7%) | 6 (3.5%) | 4 (10.3%) | |
| | **Foot care education topics taught (0–14): mean ± SD (median; IQR)** | 5.26 ± 4.67 (5; 1–9) | 1 (1; 1–1) | 3.40 ± 3.96 (1; 0–7) | 5.44 ± 4.73 (5; 1–9) | 8.51 ± 4.02 (9; 5–11) | <0.001 |

age group was 50–59 years (30.7%), with only 1.7% aged ≤19 and 16.3% aged ≥60. Over three-quarters (76.0%) had type 2 diabetes (n = 228) and 24.0% had type 1 diabetes. Diabetes duration ranged from less than 5 years for 26.3% of participants to more than 15 years for 29.0%. Only 4.0% were managing diabetes with diet alone; 49.3% were on oral medications only, 21.0% on insulin only, and 25.7% on combined oral agents and insulin. Almost 86% reported at least one other chronic condition (commonly hypertension in 56% and hyperlipidemia in 50%). Health insurance coverage was not recorded.

Education levels were low: 8.7% had no formal schooling and only 23.7% had any university education. The largest employment categories were unemployed/homemakers (45.0%) and formally employed (27.7%); 71.7% had not seen a podiatrist in the past year. Low household income (≤2,000 ILS) was reported by 37.0%, whereas 7.0% reported high income (>4,000 ILS). Table 1 details characteristics by foot self-care adherence; no differences across adherence groups were detected for sex, age group, employment, income, or diabetes type after multiple-comparison adjustment (all adjusted p > 0.20). Diabetes complications and risk factors were common: neuropathy had been diagnosed in 39.3% (20.3% unsure), prior foot ulcer in 18.7% (3.3% unsure), and amputation in 6.7% (mostly toes; 1.7% unsure). Foot deformities affected 43.3% (n = 130), most commonly other painful bumps (28.7%) and corns/calluses (18.0%); 28.7% could not reach or see the soles. Current smoking was 22.0% (66 current, 43 former). Complication rates and smoking status did not differ by adherence category after false discovery rate adjustment (all p > 0.05).

## Foot self care practices and scale reliability

Adherence to foot self-care behaviors was generally suboptimal. Only one-third of participants (33.1%) reported inspecting the bottoms of their feet daily, while 22.0% never did so. Although most participants (80.9%) washed their feet daily, fewer checked or dried between the toes (44.2% and 45.1%, respectively). Fewer than half (47.3%) always tested water temperature before immersing their feet, and only 40.3% checked inside their shoes daily. Several risky behaviors were also common: 28.6% reported soaking their feet at least occasionally (including 13.1% who soaked daily), 20.2% walked barefoot at least sometimes, and 15.9% wore shoes without socks at least occasionally Frequencies of individual foot self-care behaviors are presented in Fig 1. The majority (73.0%) cut their own toenails, usually on a monthly or more frequent basis, with 61.3% trimming about once per week. Overall, only 13.1% of participants reported engaging in all recommended foot self-care behaviors on a daily basis. Total foot self-care adherence scores ranged from 11 to 39 out of a possible 44, with higher scores indicating better self-care (Table 2).

The 11-item foot self-care scale demonstrated acceptable internal consistency (Cronbach's alpha = 0.81), supporting use of the summed composite score (0–44) as the (Table 3) primary adherence outcome. Item-level frequencies are presented in Table 2.

Data were factorable (KMO = 0.815; Bartlett's $\chi^2$(55) = 1000.5, p < 0.001). Exploratory factor analysis suggested multidimensionality: Factor 1 eigenvalue 4.07 (37% variance) and Factor 2 eigenvalue 1.42, with all negatively worded/avoidant items loading opposite to recommended behaviors consistent with two domains (preventive vs behaviors to avoid). The total score had median 27 (IQR 21–31); Shapiro–Wilk p = 0.001 with mild skewness (−0.15) and platykurtosis (kurtosis −0.76). We therefore treated the score as approximately continuous, using non-parametric tests where appropriate, and retained the composite score as the primary outcome, supported by acceptable internal consistency, The distribution of total foot self-care adherence scores is shown in Fig 2.

## Bivariable associations

Unadjusted comparisons (Table 1) showed little association between sociodemographic and adherence: sex (female 26.5 ± 6.2 vs male 26.1 ± 6.0, p = 0.65) and age (median 27 in <50 vs ≥ 50; p_trend = 0.86) were null. Education showed a weak trend (overall p = 0.046; ~23.5 with no schooling vs ~27–28 with higher education) that lost significance after false-discovery-rate control (adjusted p > 0.20). Employment and income were not significant (raw p ≈ 0.06–0.08; e.g.,

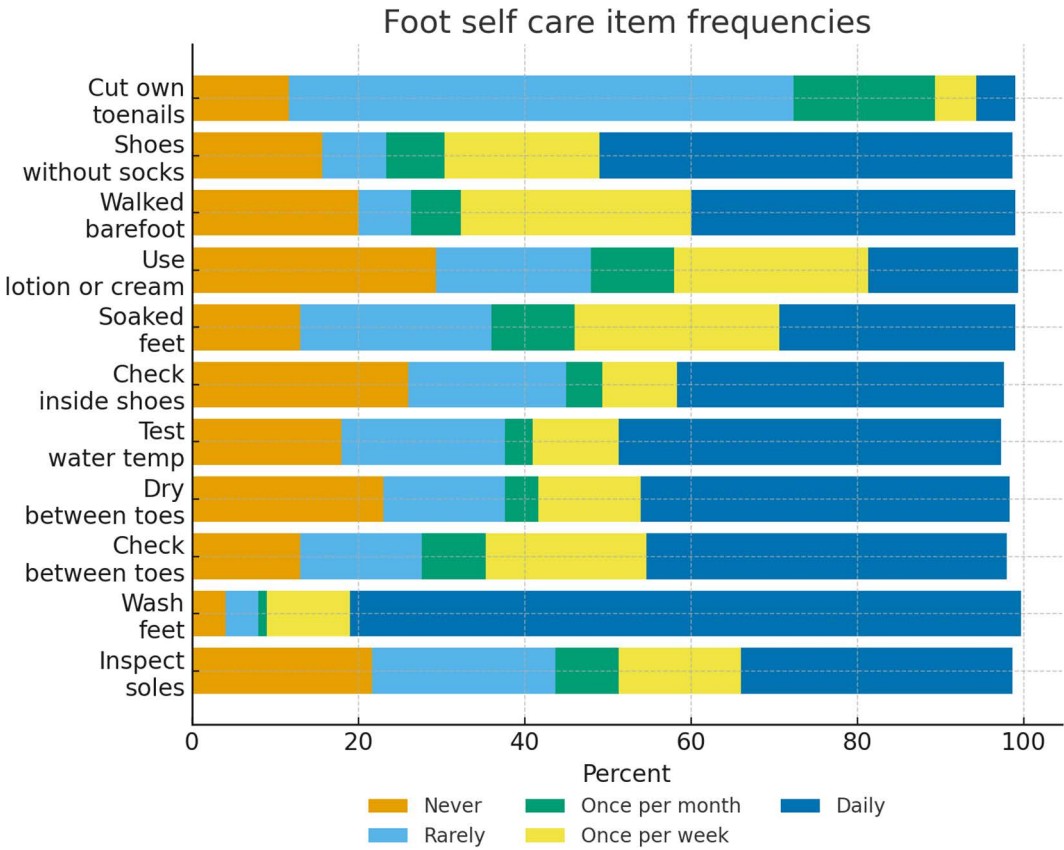

**Fig 1. Frequencies of individual foot self-care practices.**

good adherence 15.4% high-income vs 13.5% low-income, p = 0.075). Smoking status was similar (median 26 current vs 28 never, p = 0.067).

Clinical factors showed minimal bivariate links: insulin use (median 27 vs 26, p = 0.35), neuropathy (27 vs 27, p = 0.74), prior ulcer (28 vs 27, p = 0.052), and amputation (29 vs 27, p = 0.057) were non-significant after multiple-comparison adjustment (all adjusted p > 0.10). Diabetes duration correlated modestly with better scores (Spearman r = 0.15, p = 0.008; mean 27.4 for ≥15 years vs 25.9 for <5 years). The strongest associations were with exposure to foot-care education (r = 0.40, p < 0.001; median 10 topics in good vs 1 in poor adherers) and provider foot examinations (r = 0.26, p < 0.001; 3.3/7 in good vs 1.8/7 in poor), both remaining significant after FDR adjustment (adjusted p < 0.001). Age was uncorrelated (r ≈ −0.02, p = 0.75) and neuropathy symptom burden showed no significant relation (r = −0.09, p = 0.13).

### Multivariable analysis

In multivariable linear regression (44-point score), the model explained modest variance ($R^2 = 0.190$; adjusted $R^2 = 0.091$). Lower education and higher income independently predicted adherence: no formal schooling (B = −4.05, 95% CI −8.08 to −0.02, p = 0.049) and some primary schooling (B = −3.76, −6.64 to −0.88, p = 0.010) scored lower than undergraduate, while high income improved scores (B = +3.91 vs middle, +0.42 to +7.39, p = 0.028); low income was not significant (B = −0.42, p = 0.67). Adjusted predictors of foot self-care adherence are summarized in Fig 3. Longer diabetes duration predicted higher scores (B = +1.02 per category, +0.07 to +1.98, p = 0.035). Age groups were not significant (all p > 0.10).

**Table 2. Item-level foot self-care frequency and reliability statistics.**

| Self-care item | Never | Rarely | Once/ month | Once/week | Daily | Mean±SD | Item-total corr. | Alpha if dropped |
|---|---|---|---|---|---|---|---|---|
| Looked at bottoms of your feet | 65 (22.0%) | 66 (22.3%) | 23 (7.8%) | 44 (14.9%) | 98 (33.1%) | 2.14±1.59 | 0.4 | 0.259 |
| Washed your feet | 12 (4.0%) | 12 (4.0%) | 3 (1.0%) | 30 (10.0%) | 242 (80.9%) | 3.59±1.00 | 0.26 | 0.344 |
| Checked between your toes | 39 (13.3%) | 44 (15.0%) | 23 (7.8%) | 58 (19.7%) | 130 (44.2%) | 2.66±1.48 | 0.42 | 0.26 |
| Dried thoroughly between toes | 69 (23.4%) | 44 (14.9%) | 12 (4.1%) | 37 (12.5%) | 133 (45.1%) | 2.40±1.68 | 0.39 | 0.257 |
| Tested water temperature with hands before putting foot in | 54 (18.5%) | 59 (20.2%) | 10 (3.4%) | 31 (10.6%) | 138 (47.3%) | 2.48±1.63 | 0.46 | 0.229 |
| Checked shoes for foreign objects or torn linings before wear | 78 (26.6%) | 57 (19.5%) | 13 (4.4%) | 27 (9.2%) | 118 (40.3%) | 2.17±1.71 | 0.42 | 0.24 |
| Soaked your feet | 85 (28.6%) | 74 (24.9%) | 30 (10.1%) | 69 (23.2%) | 39 (13.1%) | 2.34±1.43 | −0.56 | 0.582 |
| Used moisturizing lotions or creams on your feet | 88 (29.5%) | 56 (18.8%) | 30 (10.1%) | 70 (23.5%) | 54 (18.1%) | 1.80±1.51 | 0.27 | 0.318 |
| Walked around with bare feet | 117 (39.4%) | 83 (27.9%) | 18 (6.1%) | 19 (6.4%) | 60 (20.2%) | 2.61±1.54 | −0.09 | 0.456 |
| Worn shoes without wearing any socks | 149 (50.3%) | 56 (18.9%) | 21 (7.1%) | 23 (7.8%) | 47 (15.9%) | 2.81±1.50 | 0.01 | 0.419 |
| Cut your own toenails | 14 (4.7%) | 15 (5.1%) | 51 (17.2%) | 182 (61.3%) | 35 (11.8%) | 1.30±0.91 | −0.35 | 0.476 |

**Table 3. Multivariable linear regression for total foot self-care score (0–44). Positive coefficients indicate higher self-care scores (better adherence).**

| Variable | Coefficient (B) | 95% CI | p-value |
|---|---|---|---|
| Sex (Male vs Female) | −1.97 | −4.04 to 0.10 | 0.062 |
| Type 1 diabetes (vs Type 2) | −0.85 | −3.58 to 1.89 | 0.543 |
| Insulin use (yes vs no) | 0.22 | −2.09 to 2.53 | 0.854 |
| Neuropathy (yes vs no) | −1.04 | −3.25 to 1.18 | 0.36 |
| Prior foot ulcer (yes vs no) | 1.77 | −0.96 to 4.49 | 0.203 |
| Prior amputation (yes vs no) | 2.67 | −1.38 to 6.73 | 0.197 |
| Foot deformity (yes vs no) | 0.18 | −1.74 to 2.11 | 0.854 |
| Diabetes duration (per 5-year category) | 1.02 | 0.07 to 1.98 | **0.035** |
| Education – No schooling (vs Undergraduate) | −4.05 | −8.08 to −0.02 | **0.049** |
| Education – Some primary (vs Undergraduate) | −3.76 | −6.64 to −0.88 | **0.01** |
| Education – Some secondary (vs Undergraduate) | −2.93 | −6.38 to 0.51 | 0.095 |
| Education – Some high school (vs Undergraduate) | −1.53 | −4.74 to 1.69 | 0.352 |
| Education – Completed high school (vs Undergraduate) | −1.00 | −4.64 to 2.63 | 0.589 |
| Education – Vocational (vs Undergraduate) | 0.25 | −2.91 to 3.41 | 0.876 |
| Education – Postgraduate (vs Undergraduate) | 2.28 | −4.72 to 9.27 | 0.524 |
| Income – High (vs Middle) | 3.91 | 0.42 to 7.39 | **0.028** |
| Income – Low (vs Middle) | −0.42 | −2.36 to 1.52 | 0.671 |
| Age group – ≤ 19 (vs 50–59) | 1.74 | −6.07 to 9.54 | 0.663 |
| Age group – 20–29 (vs 50–59) | −0.51 | −5.28 to 4.26 | 0.834 |
| Age group – 30–39 (vs 50–59) | −2.16 | −7.88 to 3.57 | 0.46 |
| Age group – 40–49 (vs 50–59) | −2.91 | −6.51 to 0.69 | 0.113 |
| Age group – 60–69 (vs 50–59) | −1.61 | −3.75 to 0.53 | 0.139 |
| Age group – ≥ 70 (vs 50–59) | 0.09 | −3.06 to 3.24 | 0.955 |

Model $R^2$ = 0.190 (adjusted $R^2$ = 0.091). Bold indicates $p < 0.05$.

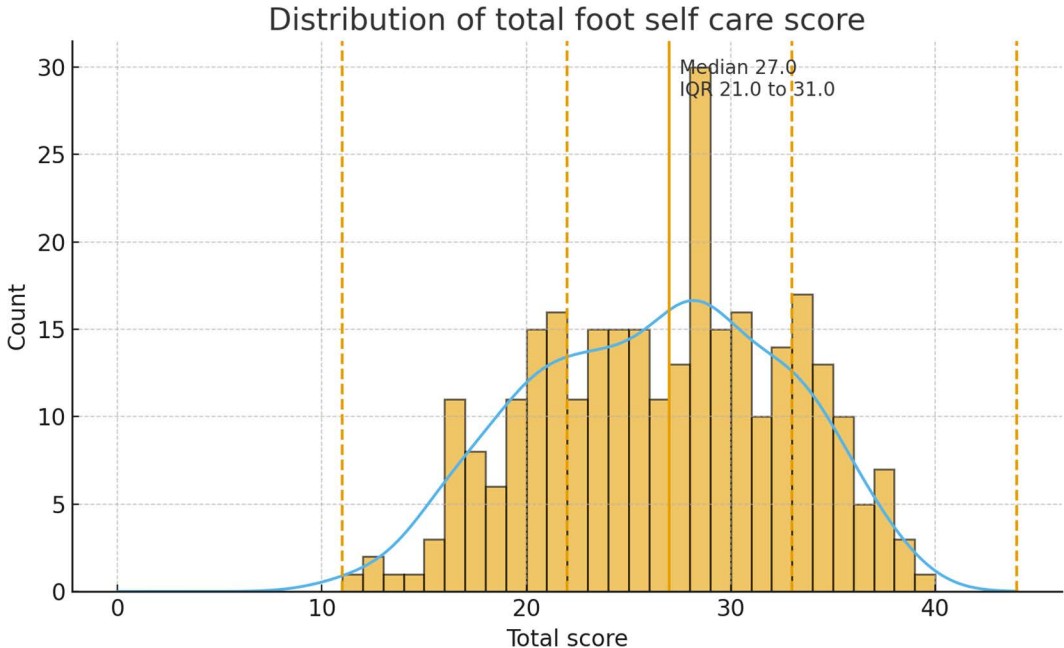

**Fig 2. Distribution of total foot self-care scores among diabetic patients.**

Male sex trended lower (B = −1.97, −4.04 to +0.10, p = 0.062). Clinical factors (type, insulin use, neuropathy, prior ulcer, amputation, deformity) were not independently associated (all adjusted p > 0.15; e.g., prior ulcer B = +1.77, −0.96 to +4.49, p = 0.20). Residual diagnostic plots did not indicate major violations of OLS assumptions; robust standard errors were therefore retained.

In the ordinal logistic model (four adherence levels), the proportional-odds assumption held (Brant test p = 0.42) and results were concordant: male sex was associated with lower adherence (OR = 0.46, 95% CI 0.24–0.87, p = 0.017); no formal education predicted lower adherence (OR = 0.26 vs undergraduate, 0.07–0.91, p = 0.036); high income increased odds (OR = 4.93 vs middle, 1.56–15.55, p = 0.006); and longer diabetes duration improved adherence (OR = 1.48 per category, 1.10–1.99, p = 0.010). Other covariates were non-significant (e.g., prior ulcer OR = 1.66, p = 0.25; insulin use OR = 0.79, p = 0.51). Model fit was acceptable (LR $\chi^2$[26] = 66.5, p < 0.001; Nagelkerke $R^2 \approx 0.21$); a partial proportional-odds specification did not improve fit.

## Discussion

This cross-sectional study provides insights into foot self-care practices among adults with diabetes in the West Bank. The findings demonstrate that adherence to recommended foot care behaviors was generally low, with only 13% of participants reporting engagement in all recommended practices on a daily basis. Although basic hygiene behaviors such as daily foot washing were widely practiced (reported by 80.9% of respondents), critical preventive measures such as daily inspection of the feet (33.1%), drying between the toes (45.1%), and daily shoe inspection (40.3%) were less frequently adopted. At the same time, a considerable proportion of patients reported engaging in potentially harmful practices, including foot soaking (28.6%), walking barefoot (20.2%), and wearing shoes without socks (15.9%). These findings indicate that while general hygiene behaviors are relatively well established, targeted preventive practices essential for reducing the risk of foot ulceration remain suboptimal.

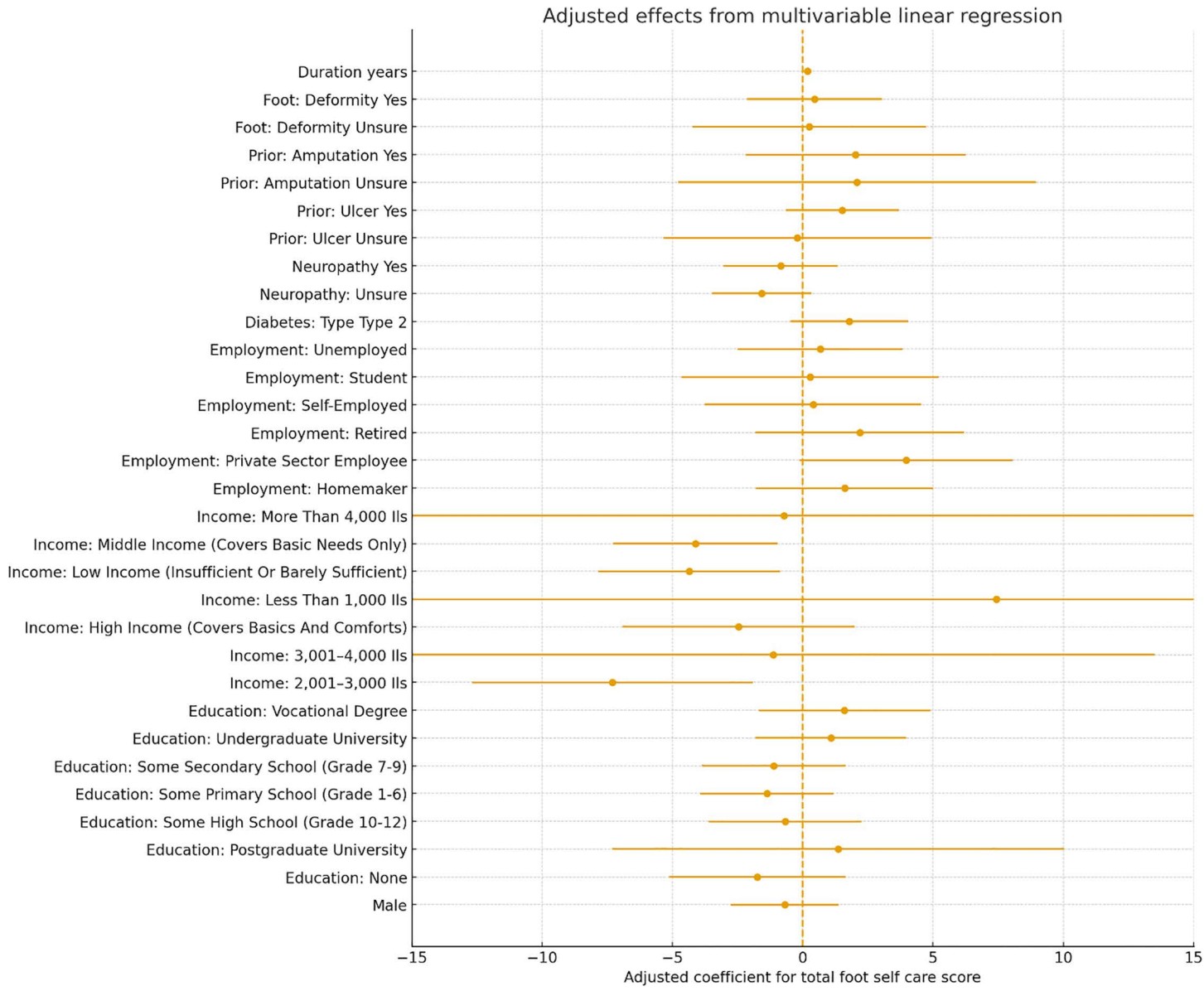

**Fig 3. Adjusted predictors of foot self-care adherence.**

The 11-item foot self-care scale demonstrated acceptable internal consistency in this study (Cronbach's alpha = 0.81), supporting use of the composite adherence score in the primary analyses. Although exploratory factor analysis suggested that recommended preventive behaviors and avoid-risk behaviors may represent related domains, the summed score provided a practical summary of overall adherence while item-level results highlight specific behavioral gaps relevant for intervention design.

Bivariate analyses indicated that sociodemographic variables such as sex, age, employment, and income showed limited associations with adherence. For example, mean foot care scores did not differ significantly between men and women (26.5 vs. 26.1, p = 0.65), nor across most age groups. However, a modest but statistically significant correlation

was observed with diabetes duration (Spearman r = 0.15, p = 0.008), suggesting that patients with longer disease duration developed slightly better foot care habits. The most striking associations emerged in relation to patient education: individuals who had been exposed to more structured foot care education topics reported significantly higher adherence scores (Spearman r = 0.40, p < 0.001). Similarly, the number of professional foot examinations was positively correlated with adherence (r = 0.26, p < 0.001), reinforcing the importance of clinical reinforcement of preventive behaviors.

In multivariable regression models, several independent predictors of adherence were identified. Educational attainment emerged as a strong determinant, with participants lacking formal schooling scoring, on average, 4 points lower than those with undergraduate education (adjusted B = −4.05, p = 0.049). Household income was also independently associated with adherence, as high-income participants scored nearly 4 points higher than middle-income participants (adjusted B = +3.91, p = 0.028). Diabetes duration remained positively associated with adherence (adjusted B = +1.02 points per 5-year interval, p = 0.035), further supporting the hypothesis that longer exposure to the disease and its management promotes improved practices. In the ordinal logistic regression, male sex was significantly associated with lower adherence (OR 0.46, p = 0.017), while higher education (OR 3.9–4.9 for university level vs. no schooling) and high income (OR 4.93, p = 0.006) substantially increased the odds of better adherence. Collectively, these analyses indicate that adherence to preventive foot care in this population is shaped by a combination of sociodemographic and experiential factors, with education and structured clinical exposure emerging as the strongest drivers of positive behavior. Taken together, the results highlight a concerning gap between recommended practices and actual patient behavior, particularly in critical preventive domains. The findings also underscore the pivotal role of health education and the influence of socioeconomic disparities on self-care adherence. These insights provide an essential foundation for targeted interventions aimed at reducing the burden of diabetic foot complications in Palestine.

The findings of this study are consistent with previous research from the Middle East and Asia, all of which have reported suboptimal adherence to recommended foot self-care among individuals with diabetes. In Oman, Al-Busaidi et al. observed that more than half of participants (54.7%) did not inspect their feet daily, despite high rates of daily washing (91.1%) and a notable prevalence of neuropathic symptoms and previous ulceration [6]. This pattern reflects the tendency, also observed in our population, for patients to maintain basic hygiene while neglecting preventive measures more directly linked to ulcer prevention.

More recent evidence from China further highlights the central role of knowledge in influencing behavior. Xie et al., in a study of 508 patients with diabetic foot ulcers, reported mean knowledge, attitude, and practice scores of 6.15/11, 35.5/60, and 26.8/50, respectively, and demonstrated through path analysis that knowledge directly and indirectly affected practice, mediated by patient attitudes [7]. Although their study population differed in disease severity, the findings converge with ours in underscoring the importance of structured education and behavioral reinforcement in improving preventive practices.

In Saudi Arabia, Alharbi and Sulaiman found that 56.5% of patients demonstrated good knowledge and 56.9% demonstrated good practices, yet only 41.5% inspected their feet daily and 40.8% walked barefoot at home [8]. Strikingly, more than two-thirds (68.5%) of participants had already developed diabetic foot complications, and fewer than 40% reported receiving foot care advice from physicians. These results are closely aligned with our findings, particularly the persistence of risky behaviors despite moderate levels of awareness, and highlight the pressing need to strengthen the role of healthcare providers in structured patient education.

Although only a minority of participants in the present study reported a history of lower limb amputation (6.7%), this figure remains clinically significant given the devastating consequences for quality of life and healthcare costs. Previous research has documented variable rates of amputation, with higher proportions generally reported in hospital-based cohorts of patients presenting with advanced ulcers, whereas lower rates are observed in community-based samples attending primary care [9]. Such variation likely reflects differences in health system capacity, access to specialized foot care, and the availability of structured education programs. The persistence of amputation cases in our outpatient

population underscores the pressing need for systematic screening, culturally adapted educational interventions, and the integration of preventive foot care into routine diabetes management to mitigate the progression toward severe outcomes.

This study shows that diabetic patients in the West Bank engage in some basic hygiene practices, such as daily foot washing, but do not consistently perform essential preventive measures like daily foot inspection, interdigital drying, or checking footwear. Risky behaviors, including barefoot walking, foot soaking, and wearing shoes without socks, were also common. These patterns indicate that patients may value general cleanliness but have limited understanding of behaviors directly linked to preventing complications.

Educational and economic factors strongly influence adherence. Patients with higher education and income were more likely to follow recommended practices, suggesting that health literacy and financial capacity play a crucial role in self-care [10]. Disease duration was also associated with better practices, which may reflect increased awareness over time as patients gain experience with managing their condition. Gender differences were evident, with men showing lower adherence, possibly due to differing health-seeking behaviors [11].

Healthcare system limitations further contribute to poor adherence. Inconsistent patient education, limited time for counseling in clinical settings, and the absence of structured diabetes education programs reduce opportunities to reinforce foot care practices [12]. The persistence of harmful behaviors despite partial awareness points to a gap between knowledge and daily practice, likely influenced by cultural habits, insufficient provider follow-up, and lack of community-based health promotion [13]. Overall, foot self-care in this population is shaped by a combination of individual knowledge, socioeconomic conditions, cultural practices, and systemic healthcare barriers. Addressing these factors in an integrated manner is essential for improving preventive behaviors and reducing the burden of diabetic foot complications.

This study has several strengths. It is among the first large-scale investigations of diabetic foot self-care practices in the West Bank, providing context-specific evidence from a region where such research is scarce. The relatively large sample size and use of a structured questionnaire allowed for a detailed assessment of foot self-care practices. Rigorous statistical analysis was employed to identify factors associated with adherence, and ethical approval with standardized data collection enhanced the reliability of the results.

Nevertheless, certain limitations should be acknowledged. The cross-sectional design prevents causal inference between patient characteristics and self-care practices. Reliance on self-reported data introduces the possibility of recall and social desirability bias. Although the scale showed acceptable internal consistency, self-reported behaviors may be affected by recall and social desirability bias. Finally, as the study was conducted in selected clinics in the West Bank, generalizability to other regions of Palestine or different healthcare contexts may be limited.

The findings carry important implications for clinical practice and health policy. Healthcare providers should integrate structured foot care education into routine diabetes management, ensuring consistent counseling, regular foot examinations, and culturally appropriate advice tailored to patients' literacy and socioeconomic conditions. Establishing multidisciplinary diabetes services with podiatric input could further strengthen preventive care [14]. At the community level, targeted health promotion initiatives are required, with particular focus on men, individuals with limited education, and those facing financial constraints. Community health workers, culturally adapted materials, and group-based education could extend support beyond clinical settings. Policymakers should incorporate standardized foot care protocols into national diabetes guidelines, invest in provider training, and expand access to affordable protective footwear [15]. Such measures would not only improve patient outcomes but also reduce the long-term economic burden associated with diabetic foot complications.

Future research should explore these issues further. Longitudinal studies are needed to track changes in self-care over time and to clarify causal relationships between education, socioeconomic status, and adherence. Intervention studies should test the effectiveness of structured education, culturally tailored health promotion, and community-based programs [16]. Qualitative research would provide deeper insight into patient beliefs, cultural practices, and provider perspectives that shape self-care behaviors. Expanding studies to rural and underserved populations would enhance generalizability

and address equity concerns [17]. Finally, economic evaluations of preventive strategies would provide valuable evidence for policymakers regarding the cost-effectiveness of investing in education and preventive interventions to reduce the burden of diabetic foot complications [18].

This study revealed that diabetic patients in the West Bank demonstrate inadequate adherence to recommended foot self-care practices, with preventive measures such as daily foot and shoe inspection performed inconsistently and harmful behaviors, including barefoot walking and foot soaking, remaining common. Adherence was influenced by educational attainment, socioeconomic status, and duration of diabetes, while systemic barriers such as limited counseling and absence of structured education further hindered effective self-care. These findings underscore the urgent need for culturally tailored education programs, greater involvement of healthcare providers in consistent counseling and screening, and policy-level support to integrate foot care into national diabetes management strategies in order to reduce complications and improve patient outcomes.

## Acknowledgments

The authors would like to thank Ola Hammad and Saja Rajabi, nutritionists, for their valuable assistance in data collection and support during the conduct of this study.

## Author contributions

**Conceptualization:** Alhareth M. Amro, salahaldeen deeb, Osama J. Makhamreh.

**Data curation:** Alhareth M. Amro, salahaldeen deeb, Bushra M. Makhamra, Lama Amro, Osama J. Makhamreh, Anas Ishqair.

**Formal analysis:** Alhareth M. Amro, salahaldeen deeb, Bushra M. Makhamra, Lama Amro, Anas K. Assi, Osama J. Makhamreh.

**Investigation:** Alhareth M. Amro, salahaldeen deeb.

**Methodology:** Alhareth M. Amro, salahaldeen deeb, Mohammad Harbi Alfrookh, Bushra M. Makhamra, Anas Ishqair.

**Project administration:** Alhareth M. Amro, salahaldeen deeb.

**Resources:** Alhareth M. Amro, salahaldeen deeb.

**Software:** Alhareth M. Amro, salahaldeen deeb, Osama J. Makhamreh.

**Supervision:** salahaldeen deeb, Afnan W. M. Jobran.

**Validation:** Alhareth M. Amro, salahaldeen deeb, Mohammad Harbi Alfrookh, Osama J. Makhamreh.

**Visualization:** salahaldeen deeb.

**Writing – original draft:** Alhareth M. Amro, salahaldeen deeb, Mohammad Harbi Alfrookh, Anas K. Assi.

**Writing – review & editing:** Alhareth M. Amro, salahaldeen deeb, Mohammad Harbi Alfrookh, Bushra M. Makhamra, Lama Amro, Anas K. Assi, Osama J. Makhamreh, Anas Ishqair, Afnan W. M. Jobran.

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
