## [Decision Letter · Decision Letter 0]

4 Jan 2026

PGPH-D-25-02859

Knowledge, Attitudes, and Practices of Foot Self-Care Among Diabetic Patients in the West Bank, Palestine: A Cross-Sectional Study

Dear Dr. Deeb,

Thank you for submitting your manuscript to PLOS Global Public Health. After careful consideration, we feel that it has merit but does not fully meet PLOS Global Public Health’s publication criteria as it currently stands. Therefore, we invite you to submit a revised version of the manuscript that addresses the points raised during the review process.

The reviewers have noted areas of significant concern that must be addressed in order to consider the manuscript for publication. I agree with these concerns and requests; please adequately ensure each of them is adequately addressed in a subsequent draft. Please try to be detailed in your "Response to Reviewers" document, indicating what changes were made in response to each request, and where in the manuscript (line numbers) these changes can be found.

We look forward to receiving your revised manuscript.

Kind regards,

Bram Wispelwey, MD, MS, MPH

Academic Editor

Journal Requirements:

1. In the ethics statement in the Methods, you have specified that verbal consent was obtained. Please provide additional details regarding how this consent was documented and witnessed, and state whether this was approved by the IRB.

Reviewers' comments:

Reviewer's Responses to Questions

**Comments to the Author**

1. Does this manuscript meet PLOS Global Public Health’s publication criteria?

Reviewer #1: Yes

Reviewer #2: Partly

Reviewer #3: Yes

2. Has the statistical analysis been performed appropriately and rigorously?

Reviewer #1: Yes

Reviewer #2: Yes

Reviewer #3: Yes

3. Have the authors made all data underlying the findings in their manuscript fully available (please refer to the Data Availability Statement at the start of the manuscript PDF file)?

Reviewer #1: Yes

Reviewer #2: Yes

Reviewer #3: Yes

4. Is the manuscript presented in an intelligible fashion and written in standard English?

Reviewer #1: Yes

Reviewer #2: Yes

Reviewer #3: Yes

Reviewer #1: Thank you for the opportunity to review this important study, which focuses on a relevant health issue in Palestine and targets an important population. However, I have a few key concerns for the authors:

1. The study title mentions "knowledge, attitude, and practice," but the results and data collection tools do not provide information on knowledge and attitude. I suggest revising the title or including those aspects in the study.

2. Regarding study participants, please detail the inclusion and exclusion criteria, including the minimum duration of diabetes required for inclusion.

3. For the study setting, please explain why these four health centers/sites were chosen and how many were available in total.

4. Concerning sampling, please provide details on the sampling framework and how the sample size was calculated.

5. For the data collection tool, include details about its development, the number of questions in each section, the original language, and any validity and reliability information.

6. Regarding the study procedure, please describe how participants were recruited, how they were contacted, and who was involved in data collection.

7. In the results section, consider adding tables to present participant details and relevant statistical information, rather than relying solely on figures, to enhance clarity.

Reviewer #2: Thank you for this paper. The topic is of significant public health importance, addressing the preventable burden of diabetic foot complications in a context where evidence is scarce. The study's large sample size (n=300) and systematic approach provide valuable data on the sub-optimal adherence to preventive care, identifying clear socioeconomic and system-level barriers. Overall, this is a valuable research article that should be published after the authors address the major methodological concerns, particularly regarding the primary outcome measure.

These are:

1. Reliability of the Foot Self-Care Scale and Primary Outcome. The most critical issue is the poor psychometric quality of the primary outcome measure, the 11-item total foot self-care score.

Low Internal Consistency: The authors report a Cronbach's alpha of 0.39 for the total scale. This value is unacceptably low for a composite score and strongly indicates that the included items do not measure a single, coherent construct ("foot self-care adherence")

Multidimensionality: The authors' own Exploratory Factor Analysis (EFA) suggested multidimensionality, with items clearly separating into two factors (preventive vs. avoidant behaviours). Furthermore, removing certain items (e.g., "soaked your feet") would significantly raise the alpha, reinforcing the lack of coherence. The Ordinary Least Squares (OLS) linear regression model used to identify independent predictors relies on this unreliable total score. The authors must revise the primary analysis.

Furthermore, I'd strongly encourages authors to make all data underlying their findings fully available and without restriction. The current statement, "The datasets used and/or analyzed during the current study are available from the corresponding author on reasonable request" is often considered insufficient unless specific legal or ethical restrictions prevent public sharing.I'd encouraged the authors to deposit their anonymized, minimal dataset (the cleaning and analysis file) into a public repository and update their DAS to provide a stable link and Digital Object Identifier (DOI). If ethical or legal restrictions exist, they must be detailed in the submission, not simply summarised as "on reasonable request"

A few additional, minor Comments:

Missing Data Imputation in Methods: The authors performed imputation for participants missing one or two foot self-care items, using the item median across the sample. Please state the percentage of participants for whom imputation was performed. While median imputation is acceptable for ordinal data, quantifying its frequency helps reviewers and readers assess potential bias.

Interpretation of Non-Significant Bivariate Findings: The Methods section describes controlling for multiple comparisons using the Benjamini Hochberg False Discovery Rate (FDR). The Results section correctly notes that several bivariate associations (e.g., education, smoking, prior ulcer) were non-significant after FDR adjustment. However, some of these variables (e.g., education) were highly significant in the final multivariable regression (p=0.049). In the Discussion section, please explicitly address this important discrepancy. The discussion should clarify that the association between these predictors (like education and income) and adherence only becomes apparent after adjusting for potential confounders in the multivariable model, thereby demonstrating the added value of the adjusted analysis.

Clarity on Normality Check: The authors report a significant Shapiro-Wilk test for the total score (p=0.001), which formally rejects the normality assumption. They justified proceeding with OLS regression by noting the mild skewness and kurtosis , and by using robust standard errors. Since OLS is the primary adjusted analysis, please confirm in the Methods or Results that the residuals of the final OLS regression model were also checked for normality and homoscedasticity. These checks are more important for validating the OLS model assumptions than the distribution of the raw outcome score alone.

Reviewer #3: Thank you for the opportunity to review this manuscript, which addresses an important and urgent problem in the developing world: the prevention of diabetic foot complications. Overall, I find the manuscript to be well written. The methodology is appropriate for the research question, and the analytical approach is clearly described.

However, the description of the sample size selection could be strengthened. While the authors state that participants were recruited consecutively, further elaboration would be helpful to better assess any potential risk of selection bias. In addition, providing more context on the usual care practices at the clinic would improve the reader’s understanding of the study setting. For example, information on how frequently patients attend the clinic and the typical patient volume would allow for a clearer assessment of the representativeness of the sample. As currently presented, it is difficult to determine whether the study population adequately reflects the broader clinic population.

The discussion section is well articulated, and the conclusions appropriately reflect the study findings.

**Do you want your identity to be public for this peer review?** For information about this choice, including consent withdrawal, please see our Privacy Policy

Reviewer #1: **Yes:** Masoud Mohammadnezhad

Reviewer #2: **Yes:** Andrew Hill

Reviewer #3: No

---

## [Decision Letter · Decision Letter 1]

2 Feb 2026

Foot Self-Care Practices Among Adults With Diabetes in the West Bank, Palestine: A Cross-Sectional Study.

PGPH-D-25-02859R1

Dear Dr. Deeb,

We are pleased to inform you that your manuscript 'Foot Self-Care Practices Among Adults With Diabetes in the West Bank, Palestine: A Cross-Sectional Study.' has been provisionally accepted for publication in PLOS Global Public Health.

Best regards,

Bram Wispelwey, MD, MS, MPH

Academic Editor

Reviewer Comments (if any, and for reference):

Reviewer's Responses to Questions

**Comments to the Author**

Reviewer #2: All comments have been addressed